# Upshaw-Schulman syndrome-associated ADAMTS13 variants possess proteolytic activity at the surface of endothelial cells and in simulated circulation

**Anton Letzer**[1☯], **Katja Lehmann**[1☯], **Christian Mess**[2], **Gesa König**[1], **Tobias Obser**[1], **Sven Peine**[3], **Sonja Schneppenheim**[4], **Ulrich Budde**[4], **Stefan W. Schneider**[2], **Reinhard Schneppenheim**[1], **Maria A. Brehm**[1] *

1 Department of Pediatric Hematology and Oncology, University Medical Center Hamburg-Eppendorf, Hamburg, Germany, 2 Department of Dermatology and Venerology, University Medical Center Hamburg-Eppendorf, Hamburg, Germany, 3 Institute for Transfusion Medicine, University Medical Center Hamburg-Eppendorf, Hamburg, Germany, 4 Haemostaseology, MEDILYS Laborgesellschaft mbH, Hamburg, Germany

☯ These authors contributed equally to this work.
* m.brehm@uke.de

**Data Availability Statement:** All relevant data are within the manuscript and its Supporting Information files.

## Abstract

ADAMTS13 regulates the hemostatic activity of von Willebrand factor (VWF). Determined by static assays, proteolytic activity <10IU/dL in patient plasma, in absence of ADAMTS13 autoantibodies, indicates Upshaw-Schulman syndrome (USS); the congenital form of Thrombotic Thrombocytopenic Purpura (TTP). We have recently functionally characterized sixteen USS-associated ADAMTS13 missense variants under static conditions. Here, we used two assays under shear flow conditions to analyze the activity of those seven mutants with sufficiently high residual secretion plus two newly identified variants. One assay determines cleavage of VWF strings bound to the surface of endothelial cells. The other, light transmission aggregometry-based assay, mimics degradation of VWF-platelet complexes, which are likely to be present in the circulation during TTP bouts. We found that 100 ng/ml of all variants were able to cleave about 80–90% of VWF strings even though 5 out of 9 exhibited activity ≤1% in the state-of-the-art static assay at the same concentration. These data indicate underestimation of ADAMTS13 activity by the used static assay. In simulated circulation, two variants, with missense mutations in the vicinity of the catalytic domain, exhibited only minor residual activity while all other variants were able to effectively break down VWF-platelet complexes. In both assays, significant proteolytic activity could be observed down to 100 ng/ml ADAMTS13. It is thus intriguing to postulate that most variants would have ample activity if secretion of 10% of normal plasma levels could be achieved.

## Introduction

ADAMTS13 (a disintegrin-like and metalloproteinase with thrombospondin type 1 motif, member 13 (OMIM #604134)) is a protease highly specific for cleavage of von Willebrand

**Funding:** This study was financially supported by research funding from the German Research Foundation (DFG, https://www.dfg.de) to the Research Group FOR1543: "Shear flow regulation of hemostasis - bridging the gap between nanomechanics and clinical presentation" (grant recipient: RS, grant number: SCHN325/7-2). The Medilys Laborgesellschaft (https://www.medilys.de) provided support in the form of salaries for authors SS and UB. The specific roles of these authors are articulated in the 'author contributions' section. The funders did not have any role in the study design, data collection and analysis, decision to publish, or preparation of the manuscript. There was no additional external funding received for this study.

**Competing interests:** The Medilys Laborgesellschaft (https://www.medilys.de) provided support in the form of salaries for authors SS and UB. This does not alter our adherence to PLOS ONE policies on sharing data and materials. The other authors declare no competing interests.

factor (VWF) [1–5]. Upon vessel injury, this large multimeric plasma glycoprotein is presented on the surface of endothelial cells (EC's) or binds to damage-exposed subendothelium and is subsequently force-activated to recruit platelets to the lesion. Platelet binding increases the tensile force along the multimers leading to unfolding of A2 domains, which harbor the ADAMTS13 cleavage site between amino acid (aa) residues Tyr1605 and Met1606 [6–8].

This size regulation of the hemostatically most active high molecular weight multimers (HMWM) of VWF is essential to prevent vessel occlusion. Thus, deficiency of ADAMTS13 leads to Thrombotic Thrombocytopenic Purpura (TTP) [9], which is a thrombotic microangiopathy caused either by the development of autoantibodies against ADAMTS13 (acquired TTP) or by mutations in the *ADAMTS13* gene. The latter form is called congenital TTP or Upshaw-Schulman syndrome (USS).

The characteristic symptoms of USS and TTP include microvascular thrombosis, tissue ischemia, and infarction [10]. Consequently, patients develop profound thrombocytopenia, severe hemolytic anemia, neurological impairment, cardiac insufficiency, renal injury, abundant schistocytes, and fever [9].

USS patients are usually treated by infusion of fresh-frozen plasma (FFP) since removal of autoantibodies by plasmapheresis is not required. In severe cases of USS prophylactic FFP infusions are necessary which impair the patients' quality of life. Disease onset and the patients' clinical courses exhibit considerable heterogeneity, even among patients with the same mutation [11]. The development of an active thrombotic microangiopathy thus seems to be triggered by additional events including but not limited to pregnancy [12–15], infections or surgery [16–18].

More than 80 USS-causing ADAMTS13 mutations, distributed throughout the entire *ADAMTS13* gene, have previously been reported [3,19–29]. We have recently identified the underlying ADAMTS13 mutations in 30 patients with USS and investigated a potential genotype/phenotype correlation by comprehensive expression studies. We reported the *in vitro* characterization of a set of 31 variants under static conditions, 10 of which harbored novel, previously undescribed, mutations [30]. Here, we performed functional analysis of ADAMTS13 variants under flow conditions. To this end, we chose those seven variants with sufficient residual secretion described in the abovementioned study [30] as well as two additional variants. Two assays were employed: One measuring VWF cleavage at the surface of endothelial cells [31], the other simulating the cleavage of VWF-platelet complexes in circulation [32,33], which might be comparable to complexes that could be formed in a TTP scenario. Our surface shear flow assay measures cleavage of VWF strings secreted by histamine-stimulated Human Umbilical Vein Endothelial Cells (HUVEC). String detection is performed using GPIbα-beads since handling is more comfortable compared to using platelets. Details on assay performance and validation as well as its feasibility for the investigation of plasma ADAMTS13 were previously described [31].

Our data show that all investigated ADAMTS13 variants exhibit residual activity when exposed to shear forces, even though half of them have no detectable activity in the state-of-the-art static assay.

## Materials and methods

### ADAMTS13 mutations

Variants p.Leu232Gln p.Asp235Tyr, p.Arg349Cys, p.Pro353Leu, p.Cys400Arg, p.Pro671Leu, p.Gly702Arg were previously identified in USS patients and characterized under static conditions [30]. Variant p.Ile222Thr was newly identified in a patient compound-heterozygously with the nonsense mutation p.Cys1275X. Variant p.Cys758Arg was previously found in a

French cohort [29] and in one of our patients, compound-heterozygously with the common duplication c.4143dupA.

## Recombinant human ADAMTS13 variants and wildtype VWF

Recombinant wildtype (wt) ADAMTS13 and VWF were derived from ADAMTS13 and VWF expression vectors, respectively, as previously described [34,35]. *In vitro* mutagenesis of the *ADAMTS13* cDNA was performed using the QuikChange® Multi Site-Directed Mutagenesis Kit (Agilent). The expression vectors containing variants of the ADAMTS13 cDNA were sequenced and then used to transform Match1T1 supercompetent cells (Thermo Fisher Scientific). Four μg of purified vector DNA were used to transiently transfect $2x10^6$ HEK293 cells (ATCC) employing Lipofectamine 2000 (Thermo Fisher Scientific) according to the manufacturer's instructions. The cells were selected for stable expression for 2 weeks by adding 500 μg/ml G418 (Thermo Fisher Scientific) to the Dulbecco modified Eagle medium (Thermo Fisher Scientific) with 10% [vol/vol] fetal bovine serum (Thermo Fisher Scientific) and 1% penicillin/streptavidin (Thermo Fisher Scientific). Seventy-two hours before harvesting the ADAMTS13 and VWF proteins from the cell supernatant, the cells were washed with PBS and the medium was exchanged with serum-free OPTIPRO-SFM medium (Thermo Fisher Scientific). Secreted VWF and ADAMTS13 variants were concentrated by centrifuge filtration using Amicon Ultra-50 NMWL with a 100 kDa and 30 kDa cutoff (Merck Chemicals), respectively. All ADAMTS13 variants were concentrated 20-fold. Afterwards the yielded concentration was determined by the Imubind® ADAMTS13 ELISA (Sekisui Diagnostics). To yield a concentration >3000 ng/ml for all mutants, the required factor for further concentration was estimated for low expressing variants, which then underwent one additional concentration step. In total, low expressing variants were concentrated between 50 to 200-fold.

The wtVWF secreted from HEK293 cells exhibits a multimer distribution comparable to the one found in pooled control plasma with some additional ultralarge multimers (see **S1 Fig**). The latter are also a typical sign of TTP in patient plasma.

## Protein quantification and activity measurement (static)

To measure ADAMTS13 activity under static conditions, the TECHNOZYM® ADAMTS-13 Activity ELISA Kit (technoclone) was employed according to the manufacturer's instructions. All samples were measured in duplicates. VWF concentration was determined by VWF:Ag-ELISA as previously described [36]. In brief, 96-well-microtiter plates were coated overnight at 4˚C with polyclonal rabbit anti-VWF (Dako, catalogue number P0082, lot number 20051014, public identifier RRID:AB_2315602, dilution 1:1000 in 50 mM carbonate buffer pH 9.6). After this and each following incubation step (1h at 37˚C), the wells are washed three times with wash buffer (0.1% BSA in PBS). 1st step: triplicates of VWF samples as well as pooled human plasma, serving as standard, diluted in PBS containing 5% BSA. 2nd step: polyclonal rabbit anti-human VWF-HRP (DAKO, catalogue number P022602, lot number 20046824, public identifier RRID:AB_579516, dilution 1:2000 in 1% BSA in PBS). Then the HRP substrate TMB (liquid substrate system for ELISA, Sigma Aldrich) was added, incubated at RT for 5 min and the reaction was stopped with 0.5 M sulfuric acid and read at 450 nm using amicroplate reader htIII (anthos).

## Western blotting

ADAMTS13 variants were stably expressed in HEK293 cells (ATCC). Seventy-two hours after changing to serum-free medium, the supernatant was harvested, concentrated 20-fold and equal volumes were analyzed by Western blotting. To determine ADAMTS13 expression, cell

lysates were produced using the MPER reagent (Thermo Fisher Scientific) according to the manufacturer's instructions. Total protein concentration was measured using the Qubit Fluorometer (Thermo Fisher Scientific). For each sample 45 μg of total protein was separated by SDS gel-electrophoresis and ADAMTS13 was analyzed by Western blotting as previously described [37]. Antibodies used were rabbit polyclonal antibody raised against aa 1128–1427 of human ADAMTS13 (Santa Cruz Biotechnology, catalogue number sc-25584, clone H-300, lot number C0504, public identifier RRID:AB_2222029, dilution 1:2000) and HRP-conjugated polyclonal goat anti-rabbit (DAKO, catalogue number P0448, lot number 94764, public identifier RRID: AB_2617138, dilution 1:2500). For the loading control a monoclonal mouse anti-β-actin antibody (Sigma Aldrich, immunogen: slightly modified β-cytoplasmic actin N-terminal peptide, Ac-Asp-Asp-Asp-Ile-Ala-Ala-Leu-Val-Ile-Asp-Asn-Gly-Ser-Gly-Lys, conjugated to KLH, catalogue number A5441, clone Ac15, lot number C0504, public identifier RRID:AB_476744, dilution 1:5000) was used with a secondary HRP-conjugated polyclonal goat anti-mouse antibody (DAKO, catalogue number P0447, lot number 71312, RRID:AB_2617137, dilution 1:2500).

## Surface shear flow assay

Channel μ-slides I$^{0.4}$ ibidi treat® (ibidi) were coated with 2% gelatine (w/v) in PBS for 1 h at RT and washed thrice with PBS. Then $1.45 * 10^6$ single donor Human Umbilical Vein Endothelial Cells (HUVEC, Promocell) were seeded in the μ-slides in Endothelial Cell Growth Medium (Promocell) supplemented with 1 mM $MgSO_4$ and fetal bovine serum (Thermo Fisher Scientific, final concentration 5% (v/v)) and cultured overnight at 37°C and 5% $CO_2$ in a humidified chamber (OLAF, ibidi). Then, the shear-flow assays were performed as previously described by us [31]. In brief, the cells were washed with 1 ml pre-warmed (37°C) HBRS buffer (140 mM NaCl, 5 mM KCl, 1 mM $MgCl_2$, 1 mM $CaCl_2$, 10 mM HEPES, pH 7.4) and the channel slide was connected to a perfusion set air bubble-free and mounted in an incubation unit (Tokai-Hit, 5% $CO_2$, 37°C) within a fluorescence microscope (BZ-9000, Keyence). The cells were perfused at 5 dyne/cm$^2$ shear stress with 14 ml HBRS buffer containing 100 μl GPIb-beads (INNOVANCE® VWF Ac assay kit, Siemens Healthcare Diagnostics) and 1 mM histamine (Sigma Aldrich). VWF string formation, detected by binding of GPIbα-beads to VWF, was observed using a Plan Fluor ELWD DM x20 phase contrast objective with a numerical aperture of 0.45. To determine catalytic activity of recombinant ADAMTS13 variants, the proteins (pre-incubated at 37°C for 10 min) were added to one of the reservoirs in a final concentration of 100 ng/ml. In control measurements, no ADAMTS13 was added. VWF string cleavage was monitored by capturing of an automated time-lapse multi-picture series of 12 images every 36 sec for 12 min, starting before addition of ADAMTS13 variants. In brief, the 12 images of each time point were merged to one image and the length of 100 strings (in mm) was measured using the ImageJ software [38,39]. The overall string length at 0 min was defined as 0**%** cleavage and % cleavage for each frame was determined by the decrease of the overall string length in %. Details on assay performance and validation as well as detailed description of data analysis was previously described [31].

Every experiment was performed at least 3 times. Data analysis was performed using the GraphPad Prism software version 5.02 for Windows, GraphPad Software, La Jolla California USA, www.graphpad.com.

## VWF-platelet complex degradation by ADAMTS13 using a light transmission aggregometer (LTA)

For this study, platelets were isolated from only residual amounts of anonymous peripheral blood samples, which were routinely taken from healthy blood donors at the Institute of

Transfusion Medicine, University Medical Center Hamburg-Eppendorf (Hamburg, Germany) and would have otherwise been discarded. The blood donors gave their general written consent to use their blood samples for scientific studies in an anonymized form. The samples were split in two 50 ml tubes and diluted with modified calcium-free Tyrode's buffer (137 mM NaCl, 2.7 mM KCl, 0.48 mM NaH$_2$PO$_4$, 2.7 mM glucose, 5 mM Hepes, pH 6.5) to a final volume of 50 ml and apyrase was added to a final concentration of 0.65 U/ml. All samples were centrifuged at 1590 xg for 15 min at room temperature (RT) with reduced deceleration (acceleration: 9; brake: 4). After aspiration of the clear buffer fraction, it was replaced with modified calcium-free Tyrode's buffer to 50 ml in each tube. In the following two wash steps, the concentration of apyrase was reduced by 50% each time; the last wash contained no apyrase. Centrifugation was performed as above. After the third wash, the samples were adjusted to 35 ml with platelet resuspension buffer (modified calcium free Tyrode's buffer containing 5% bovine serum albumin, pH 7.4), centrifuged for 7 min at 490 xg and the platelet-rich buffer fraction was separated from the hematocrit. The wash protocol was modified from [40].

To determine cleavage of VWF-platelet complexes, we employed a modified agglutination assay using a light transmission aggregometer (LTA), which has previously been described by Denorme et al. [32,33]. For our application we adjusted platelet and ADAMTS13 concentrations. The platelets were diluted to 300*10$^3$ cells/µl in resuspension buffer, in a glass cuvette containing a stir bar, and placed in the LTA. After starting to record the turbidity and setting the baseline, recombinant wtVWF and Ristocetin were added to final concentrations of 10 µg/ml and 0.6 mg/ml, respectively. After 10 min, ADAMTS13 variants were added (final concentration 1000 ng/ml) and turbidity was recorded for additional 50 min. Every experiment was performed at least 3 times.

To investigate the sensitivity of the assay and the minimal ADAMTS13 concentration required for sufficient complex degradation, different concentrations of recombinant wtADAMTS13 were added ranging from 0 to 1000 ng/ml.

ANOVA analysis and statistical post hoc tests were conducted in *R* version 3.5.1 [41] and changepoint analysis was performed via binary segmentation implemented in the *R* package changepoint [42]. The raw signal of each curve was sampled at two points in time (ADAMTS13 variants at 8 min and 50 min, different concentrations of wtADAMTS13 at 7 and 45 min) by calculating the intra-changepoint mean at this point. The difference between these two means quantifies the degradation of VWF-platelet complexes by ADAMTS13 variants.

## Results

### Secretion of investigated ADAMTS13 variants

Goal of this study was to investigate the residual proteolytic activity of nine USS-associated ADAMTS13 variants under shear flow conditions. As shown in **Fig 1A**, p.Ile222Thr, p.Leu232Gln and p.Asp235Tyr are located in the metalloprotease domain (MP), p.Arg349Cys and p.Pro353Leu in the disintegrin-like (Dis) domain, p.Cys400Arg in the thrombospondin type 1 (TSP) repeat number 1 (1), p.Pro671Leu in the Spacer domain and p.Gly702Arg and p.Cys758Arg in TSP 2 and 3, respectively.

Secretion defects have been described as the major cause of loss of ADAMTS13 activity in the plasma [23,43–45]. We did not previously investigate the secretion of variants p.Ile222Thr and p.Cys758Arg and the other recombinant mutants were only described with respect to expression and secretion after transient transfection [37]. Since we have established stably expressing HEK293 cell lines for this study, we determined expression and secretion of all variants by Western blotting. When corrected for the β-actin loading control (**Fig 1B, lower**

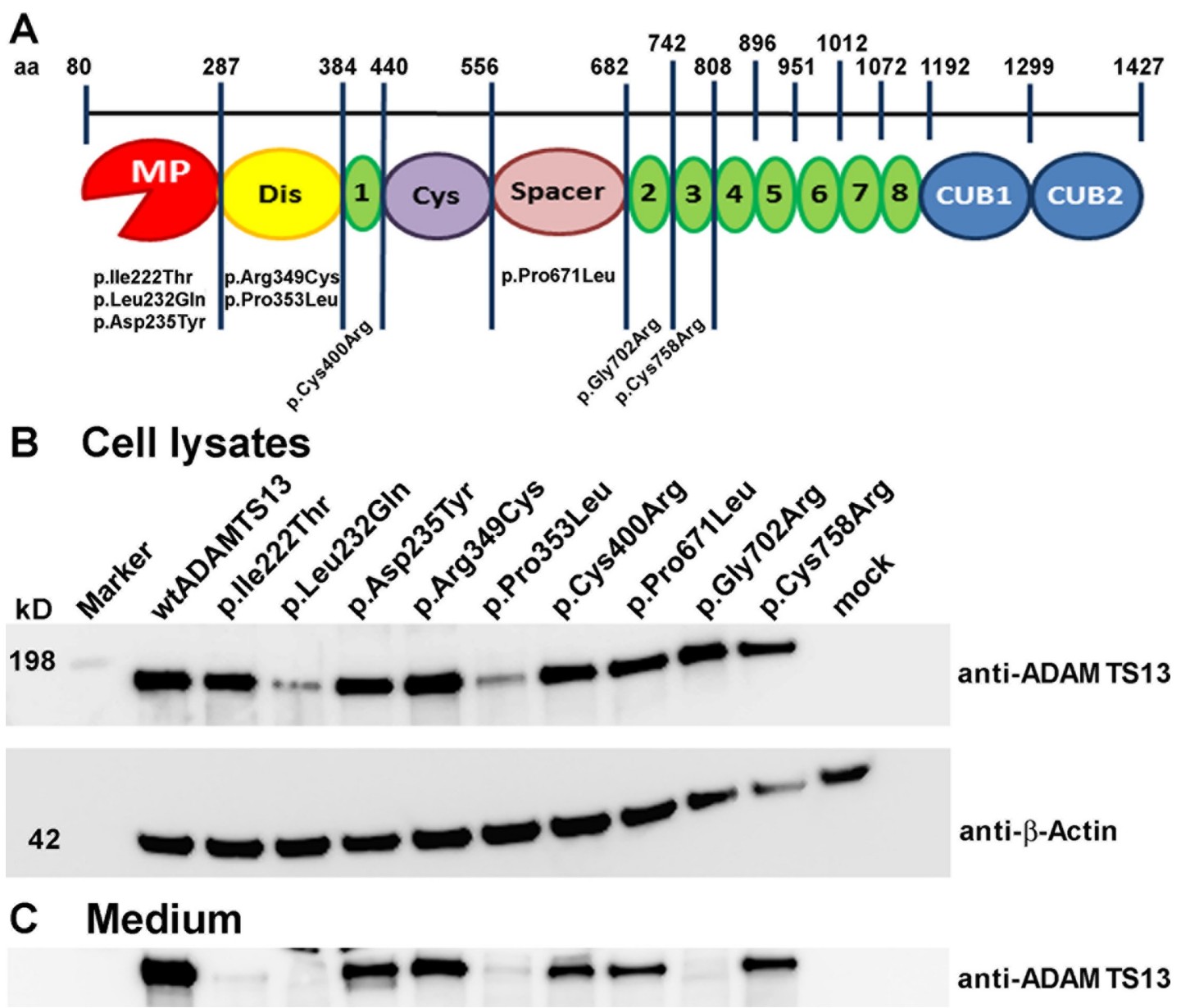

**Fig 1. Location of ADAMTS13 mutations in the protein and variant expression and secretion in HEK293 cells. (A)** Schematic presentation of ADAMTS13 domains with indicated amino acid (aa) boundaries. Metalloprotease domain (MP, red), disintegrin-like domain (Dis; yellow), TSP repeats (1–8; green), cysteine-rich domain (Cys; violet), Spacer domain (pink), and CUB domains (blue). Investigated variants are shown below the domain in which the respective missense mutation is located. **(B,C)** ADAMTS13 variants were stably expressed in HEK293 cells. Seventy-two hours after changing to serum-free medium, the supernatant was harvested, concentrated 20-fold, and equal volumes were analyzed. To determine ADAMTS13 expression, cell lysates were produced. The intracellular **(B)** and extracellular **(C)** proteins were separated by SDS-PAGE and analyzed for ADAMTS13 employing a rabbit anti-ADAMTS13 antibody and a secondary HRP-coupled goat anti-rabbit antibody. As loading control, β-actin was detected **(B)**.

**panel)**, densitometric analysis of the cell lysate Western blots showed that only variants p. Leu232Gln and p.Pro353Leu exhibited much lower levels of intracellular protein compared to wtADAMTS13 indicating reduced expression or enhanced recycling of misfolded protein. Western blot analysis of the medium and desitometric comparison of the wtADAMTS13 band with the mutant bands further revealed that all variants exhibited reduced secretion compared to wtADAMTS13 (set to 100% in densitometric analysis) (**Fig 1C**). Groups of variants with

mild, strong and severe secretion defects could be identified: While p.Asp235Tyr and p. Arg349Cys are still fairly well secreted with 24% and 40%, respectively, a stronger secretion defect was observed for variants p.Pro353Leu (12%), p.Cys400Arg (14%), p.Pro671Leu (11%) and p.Cys758Arg (14%). Very low residual secretion was found for variants p.Ile222Thr (5%), p.Leu232Gln (<1%) and p.Gly702Arg (5%) (**Table 1, second column**).

## Proteolytic activity of ADAMTS13 variants under static conditions

Often, the activity of recombinant ADAMTS13 variants is measured in x-fold concentrated medium without consideration of the actual protein concentration. Here, we wanted to investigate the catalytic activity of ADAMTS13 variants under static and flow conditions. To obtain comparable data, independent of the secretion efficiency, we always measured equal concentrations of all recombinant proteins. We have previously described that 10% of the normal ADAMTS13 plasma concentration (100 ng/ml) is the most feasible to use in our shear flow assay [31]. We thus also tested the activity of this protein concentration under static conditions employing the TECHNOZYM® ADAMTS-13 Activity ELISA Kit (**Table 1, third column**). For variants p.Ile222Thr, p.Leu232Gln, p.Asp235Tyr and p.Arg349Cys, the activity was below the detection limit, p.Pro671Leu only showed about 1% activity and p.Pro353Leu reached 16.8%. Unexpectedly, p.Cys400Arg, p.Gly702Arg as well as p.Cys758Arg exhibited high activities of 47.9%, 153.9% and 114.1%, respectively. To investigate if the same concentration of these variants exhibits a similar activity under flow conditions, we next performed two flow assays.

## Proteolytic cleavage of VWF strings by ADAMTS13 variants

We have previously described a shear flow assay to determine the proteolytic activity of ADAMTS13 towards VWF strings bound to the surface of endothelial cells. HUVEC were stimulated to release VWF strings using histamine and the strings were visualized using GPIbα-beads. Time-lapse images were recorded every 36 sec for 10.5 min to document the decrease in the total length of 100 strings after addition of 100 ng/ml ADAMTS13. Representative string analyses after 0 and 10.5 min are shown for wtADAMTS13 in the S2 Fig. Our data reveal that all variants appear to have high residual activity (**Fig 2A** and **Table 1**). Considerable differences could be observed after 2.4 min (**Fig 2B**) indicating that the initial activity and/or substrate binding are altered by some mutations. Here, mutations p.Asp235Tyr, in the metalloprotease domain, and p.Pro353Leu, in the Dis domain (**Fig 1A**), exhibited the strongest effects. Specificity of our assay for ADAMTS13 activity was confirmed in control measurements, which showed that only minor events of string detachment were observed in absence of ADAMTS13 (**Fig 2A**). These were most likely due to mechanical disruption of cell surface attachment.

## Activity of ADAMTS13 variants in simulated circulation

In vivo, the consequence of VWF string cleavage close to the surface is the release of long VWF multimers with attached platelets into the circulation. In a TTP scenario, these complexes could then lead to occlusions in the microvasculature. We have thus used a modified agglutination assay [32,33] to investigate the fate of these VWF-platelet complexes in presence of ADAMTS13 variants.

Employing a light transmission aggregometer, formation of VWF-platelet complexes was induced by addition of 0.6 mg/ml Ristocetin to washed platelets ($300^*10^3$ /µl) and 10 µg/ml recombinant wtVWF. As visualized by the decrease in turbidity, the complexes formed immediately after addition of Ristocetin (**Fig 3A**, green arrow). After 10 min, 1000 ng/ml

**Table 1. Comparison of ADAMTS13 parameters determined by static and flow assays.**

| ADAMTS13 variant | Secretion [%] | Activity of 100 ng/ml static [%] ± SD | Activity of 100 ng/ml at surface after 2.4 min [%] ± SEM | Activity of 1000 ng/ml in circulation [%] ± SEM |
|---|---|---|---|---|
| wtADAMTS13 | 100 | 100 ± 10,7 | 87 ± 5.1 | 100 ± 13.8 |
| p.Ile222Thr | 5 | < min | 78 ± 3.5 | 64 ± 13.1 |
| p.Leu232Gln | <1 | < min | 58 ± 4.3 | 38 ± 3.1 |
| p.Asp235Tyr | 24 | < min | 29 ± 4.0 | 2± 2.3 |
| p.Arg349Cys | 40 | < min | 45 ± 1.3 | 23 ± 7.8 |
| p.Pro353Leu | 12 | 16,8 ± 2,5 | 27 ± 3.0 | 59 ± 11.1 |
| p.Cys400Arg | 14 | 47,9 ± 2,0 | 77 ± 7.4 | 129 ± 11.6 |
| p.Pro671Leu | 11 | 1,1± 0,2 | 44 ± 5.4 | 109 ± 14.80 |
| p.Gly702Arg | 5 | 153,9 ± 10,4 | 80 ± 4.7* | 118 ± 15.4 |
| p.Cys758Arg | 14 | 114,1 ± 0,9 | 90 ± 0.5 | 88 ± 19.5 |

*For one protein batch, secretion was determined by Western blotting and densitometric analysis. Activity measurements under static conditions were performed using the TECHNOZYM® ADAMTS-13 Activity ELISA Kit (technoclone). All variants were measured in duplicates. Shown are mean ± SD normalized to wtADAMTS13 = 100%. Activity under flow conditions at the surface of HUVEC was compared 2.4 min after addition of ADAMTS13 variants (data shown in Fig 2B) Mean ± SEM of 3–4 experiments. \* For variant p.Gly702Arg, this value was determined in a previously published study [31]. Activity in simulated circulation was quantified by analyzing turbidity before and after addition of ADAMTS13 in LTA measurements (Fig 3). Mean ± SEM of 3–4 experiments.*

ADAMTS13 variants (100% of the normal plasma wtADAMTS13 concentration) were added (**Fig 3A**, red arrow) and turbidity was recorded for additional 50 min. The ability of respective variants to cleave VWF, is visualized by an increase in turbidity because the production of smaller complexes leads to a cloudier solution (**Fig 3A**). Changepoint analysis (examples shown in S3 Fig) determining the difference in turbidity plateaus before and after addition of ADAMTS13 variants revealed that only p.Asp235Tyr is unable to proteolyze VWF-platelet complexes. Compared to wtADAMTS13, a reduced catalytic activity was found for variants p.Ile222Tyr, p.Leu232Gln, p.Arg349Cys and p.Pro353Leu, while the activity was normal for variants p.Cys400Arg, p.Pro671Leu, p.Gly702Arg and p.Cys758. Statistically significant differences in the Wilcoxon signed-rank post hoc test were only reached for p.Asp235Tyr and p.Arg349Cys (**Fig 3B** and **Table 1**).

## Minimal required concentration of wtADAMTS13 to sufficiently reduce VWF-platelet complexes

Since the reference interval for normal ADAMTS13 concentrations is 700–1000 ng/ml, we used 1000 ng/ml for all variants in the agglutination assay. But in patients harboring USS-associated mutations, defects in protein synthesis and secretion lead to low ADAMTS13 plasma concentration. To determine, which minimal concentration of ADAMTS13 would be required to achieve sufficient degradation of VWF-platelet complexes in the circulation, we further measured additional concentrations of wtADAMTS13 ranging from 0 and 1000 ng/ml (**Fig 4A**). Changepoint analysis revealed that all concentrations ≥100 ng/ml showed a statistically significant complex size reduction (**Fig 4B**).

## Discussion

We have used two methods that allow determining the ability of ADAMTS13 variants to proteolyze VWF strings [31] and VWF-platelet complexes [32,33], importantly, under shear flow conditions. In the used assays, cleaved VWF remains bound to GPIbα-beads or platelets, rendering multimer analysis after cleavage impossible. However, the specificity of our assays for

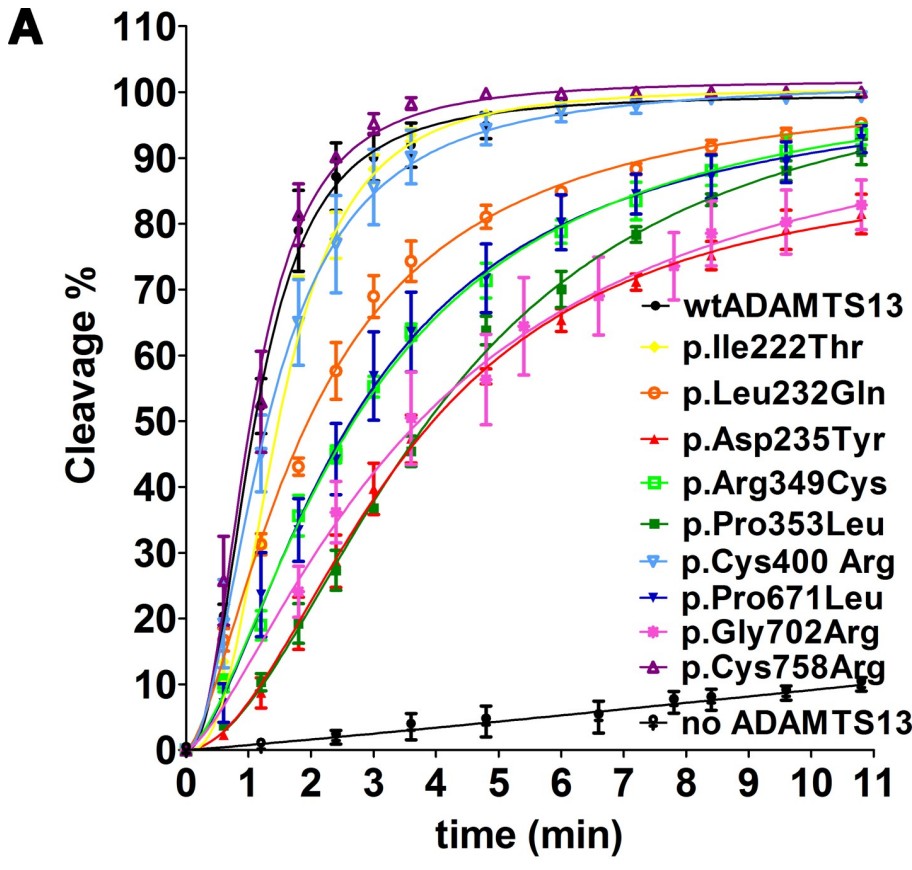

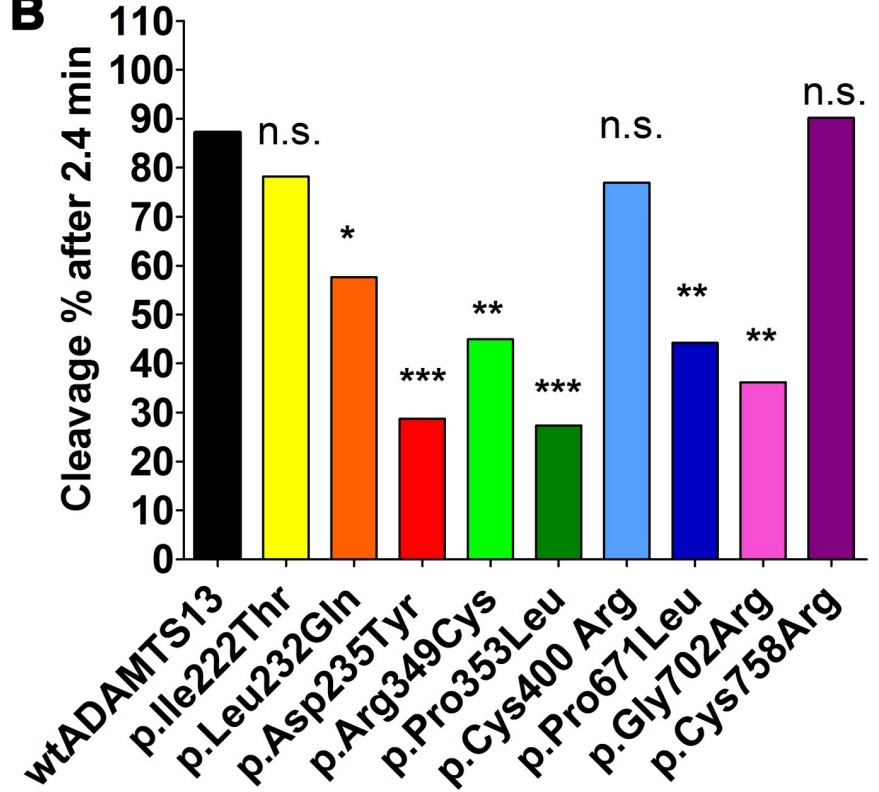

**Fig 2. Proteolysis of VWF strings at the surface of HUVEC by recombinant ADAMTS13 variants under flow.** (A) HUVEC were perfused at 5 dyne/cm$^2$ shear stress with 1 mM histamine and 100 µl of GPIbα-beads in HBRS buffer. Capturing of an automated time-lapse multi-picture series of 12 images every 36 sec for 12 min was started before addition of 100 ng/ml recombinant ADAMTS13 variants. Analysis was performed by merging the 12 images of each time point to one big image, a 100 µm scale bar was used to calibrate the scale in ImageJ to mm and the length of 100 strings was measured. The overall string length at 0 min was defined as 0**%** cleavage and % cleavage for each frame was determined by the decrease of the overall string length in %. Mean values ± SEM of 3–4 experiments per variant are shown. Exemplary analyses are shown in the **S2 Fig.** (B) % cleavage of VWF strings by ADAMTS13 variants within 2.4 min. The values were extracted from the experiments shown in panel (**A**) and are presented as bar graphs. Mean values ± SEM of 3–4 experiments. Unpaired Student's t test was performed using GraphPad Prism version 5.02 for Windows, GraphPad Software (La Jolla California USA, www.graphpad.com). ns p > 0.05, *p ≤ 0.05, **p ≤ 0.01, ***p ≤ 0.001.

ADAMTS13 is shown by the negative controls without ADAMTS13 in which only minor cleavage events were observed.

The agglutination assay employs washed platelets—to remove endogenous wtADAMTS13—and could theoretically also be used to investigate the activity of ADAMTS13 variants in patient plasma. We found that significant complex degradation requires at least the activity of 100 ng/ml wtADAMTS13. With an established reference interval of 700–1000 ng/ml ADAMTS13, this concentration corresponds to about 10% of normal plasma levels. Thus, our data are in accordance with the defined hallmark of USS: severe functional deficiency of ADAMTS13 with activity < 10 IU/dL [46,47].

Interestingly, our data show that almost all investigated ADAMTS13 variants exhibit significant residual activity when exposed to shear forces. Except for variants p.Leu232Gln, p. Asp235Tyr and p.Arg349Cys, they possess activities above 50% of ADAMTS13 under conditions mimicking cleavage in the circulation (**Table 1**). At the cell surface, all variants were even able to cleave at least 80% of VWF strings within 10 min (**Fig 2A**).

During the manually performed string measurements using the ImageJ software, we observed that a reduction of the total string length mostly originated from cleavage of whole single strings close to the cell surface. Two or three cleavage events within the same string were barely observed. Thus, only a very low activity is enough to lead to single cleavage events and subsequent significant reduction of total string length. Since a limited amount of strings is available for cleavage, which is probably lower compared to the *in vivo* situation, this observation indicates a limitation of this assay that might explain the higher residual activity of all ADAMTS13 variants in the string cleavage assay compared to the complex degradation assay, which requires multiple cleavage events to achieve an efficient complex size reduction.

In both shear flow assays, we have found that sufficient VWF proteolysis could be achieved by almost all variants at 10% of the normal plasma concentration. Our data thus confirm previous suggestions that impaired secretion is the disease-causing variable for most variants [26,48,49]. Out of the variants investigated here, only p.Asp235Tyr and p.Arg349Cys exhibit significant residual secretion but strongly reduced activity under static as well as flow conditions (**Table 1**). Since these mutations are located within or in vicinity of the catalytic domain (**Fig 1A**), these data indicate changes in the initial activity and/or altered substrate binding compared to wtADAMTS13.

This observation is in line with a study by de Groot et al., showing that mutational substitution of Arg349 in the Dis domain reduces cleavage of VWF under static conditions approximately 20-fold. Since both an increase in $K_m$ and decrease in $k_{cat}$ were observed, the authors suggest changes in both functional substrate binding and substrate turnover [50]. Crystal structures of a construct including domains Dis, TSP1, Cys and the spacer (aa 287–685) confirmed that Arg349 lies within exosite-1, which is essential for ADAMTS13 interaction with

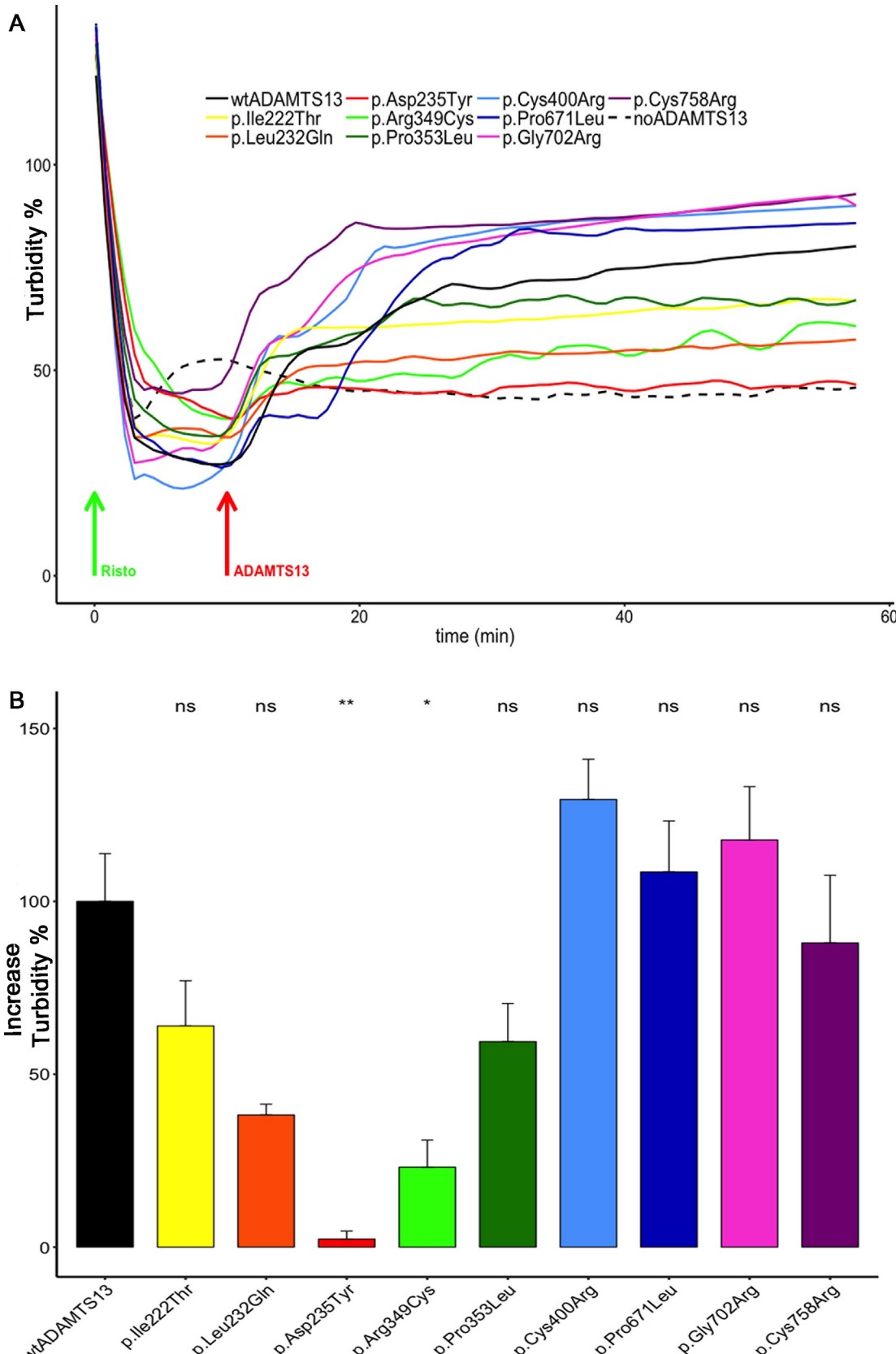

**Fig 3. Modified agglutination assay employed to measure degradation of platelet-VWF complexes by ADAMTS13 variants.**
(**A**) Washed platelets (300*10$^3$ cells / μl), in a glass cuvette containing a stir bar, were placed in the light transmission aggregometer. After starting to record the turbidity and setting the baseline, recombinant wtVWF and Ristocetin (green arrow) were added to final concentrations of 10 μg/ml and 0.6 mg/ml, respectively. After 10 min, ADAMTS13 variants were added (final concentration 1000 ng/ml) (red arrow) and turbidity was recorded for additional 50 min. Every experiment was performed at least 3 times. Simplified mean degradation curves by local regression (Loess) smoothing with neighborhood parameter 0.1 are shown. (**B**) The difference in turbidity plateaus before and after addition of ADAMTS13 variants in the LTA measurements shown in (**A**) was analyzed by Kruskal-Wallis one-way analysis of variance (p = 0.00058) with Wilcoxon signed-rank post hoc test (exemplary analyses are shown in the S3 Fig) ns p > 0.05, *p ≤ 0.05, **p ≤ 0.01, ***p ≤ 0.001). p.Asp235Tyr: p = 0.0066; p.Arg349Cys: p = 0.0168.

the VWF A2 domain [51]. Reduced secretion and activity under static conditions was also previously described for mutation p.Pro353Leu [52].

We also observed reduced secretion for all variants. Since we have performed this experiment only with one batch of stably expressed proteins, we cannot rule out minor differences in expression and secretion between different protein expressions. Nonetheless, the data shown here are in good accordance with our previous study, investigating secretion after transient transfection [30]. Compared to the stable expression shown here, then, only p.Leu232Gln exhibited higher intracellular and extracellular levels and p.Asp235Tyr showed a slightly decreased secretion. These data indicate that some differences in the extent of secretion defects can be observed between stable and transient recombinant protein expressions.

All recombinant mutants were measured at the same concentration (100 ng/ml) under static as well as flow conditions. At this level, variants p.Cys400Arg, p.Gly702Arg and p. Cys758Arg exhibited high catalytic activities even under static conditions, again indicating that low plasma concentration is the predominant cause of USS in patients carrying these mutations. Surprisingly, no or very little activity was detected for all other variants even though they exhibited significant activity under flow conditions at the same concentration (**Table 1**). These data indicate that the static assay, which currently is the most commonly used in diagnostics, can underestimate ADAMTS13 activity. This discrepancy is most likely due to the absence of shear and the use of immobilized, truncated A2 domain fragments. These fragments lack the ADAMTS13 binding sites outside of the A2 domain, which might be necessary for these mutants to make use of their residual activity. For example mutation p.Pro671Leu is located in the spacer domain, which has the highest affinity for A1 [53]. Prolines have significant impact on secondary structure and proximity of Pro671 to residues Arg660, Tyr661 and Tyr665, which are essential for VWF binding and cleavage [54,55], might significantly reduce their interaction with an isolated A2 domain as it is used in the static assay. In contrast, full-length VWF includes the D4 and C domains that provide additional binding sites for ADAMTS13 [56]. In the shear flow assays these sites present in the multimeric VWF substrate might be sufficient to support binding of p.Pro671Leu. Since its catalytic domain is most likely unaffected by the mutation, binding might be the limiting factor for cleavage by this variant. These aspects could explain the discrepancy between the results obtained by the static versus flow assays.

Using a static assay employing a soluble A2 domain fragment, Xiang et al. [57] further described complete inactivation of ADAMTS13 by mutation p.Leu232Asn. The leucine residue 232 was predicted to be a candidate residue that might interact with VWF Leu1603, but as a reason for absence of activity a massive secretion defect was also taken into consideration. Our study shows that p.Leu232Asn exhibits residual catalytic activity under flow at a concentration of 100 ng/ml and can confirm that a secretion defect is the main disease-causing mechanism underlying USS due to mutation of Leu232. Similarly, we found absence of activity under static conditions and low plasma concentration for variant p.Ile222Thr, which exhibits wildtype-like activity under flow conditions.

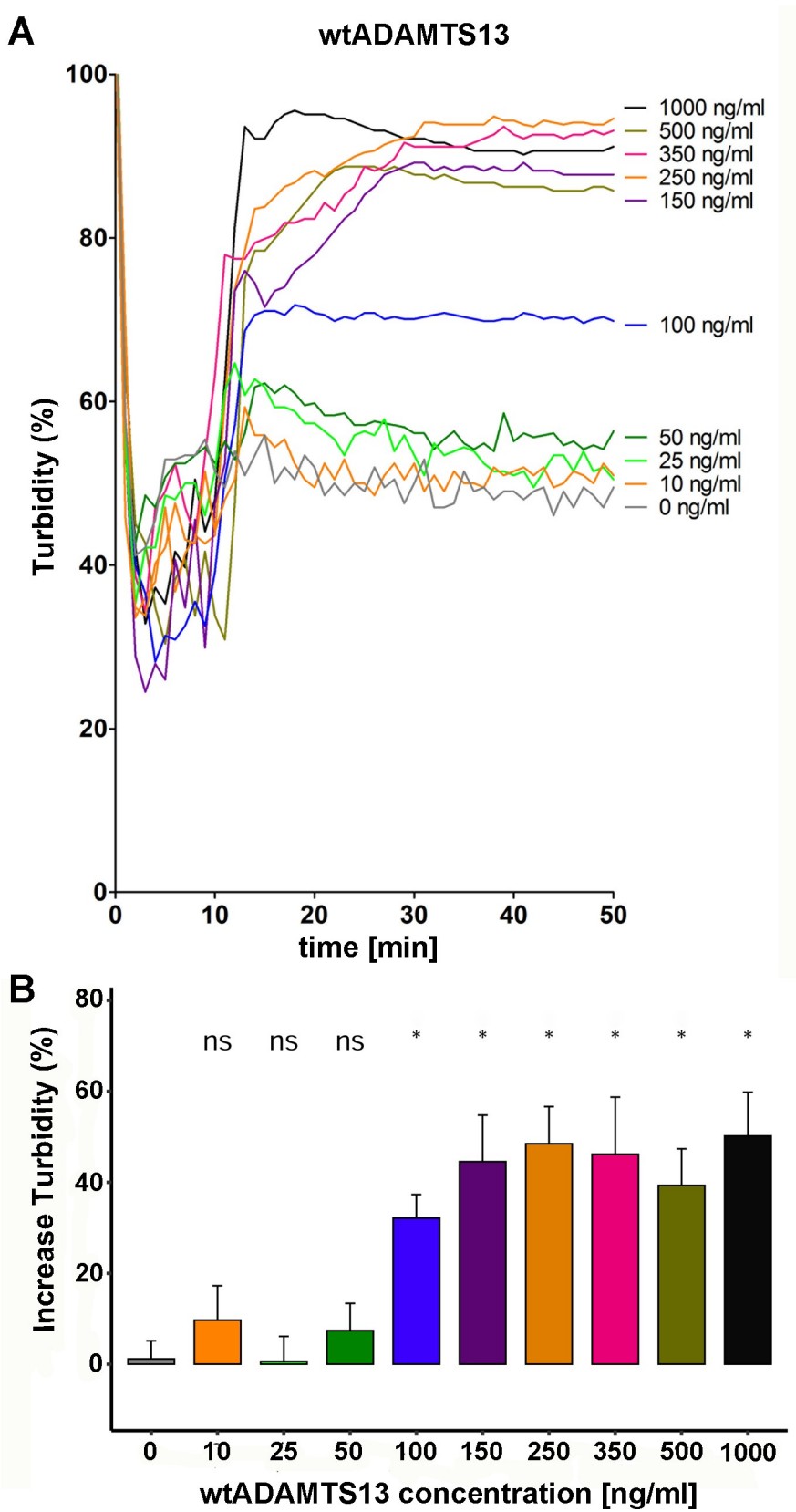

**Fig 4. Minimal concentration of ADAMTS13 required for significant proteolysis of VWF-platelet complexes in the LTA assay.** Washed platelets (300*10³ cells / μl) were placed in the light transmission aggregometer (LTA). After starting to record the turbidity and setting the baseline, recombinant wtVWF and Ristocetin were added to final concentrations of 10 μg/ml and 0.6 mg/ml, respectively. To determine the minimal concentration of ADAMTS13 required for significant proteolysis of VWF-platelet complexes in the LTA assay, different concentrations ranging from 0–1000 ng/ml were added after 10 min, and turbidity was recorded for additional 40 min. Every experiment was performed at least 3 times and flattened mean curves are shown for each concentration (0, 10, 25, 50, 100, 150, 250, 350, 500, 1000 ng/ml). **(B)** The difference in turbidity plateaus before and after addition of wtADAMTS13 in the LTA measurements shown in **(A)** was analyzed by Kruskal-Wallis one-way analysis of variance (p = 0.0023) with Wilcoxon signed-rank post hoc test. ns p > 0.05, *p ≤ 0.05.

Since we needed high concentrations of the investigated recombinant proteins, we used HEK293 cells for protein expression. Due to the used CMV promotor and the high expression capacity of these cells, it is possible that intra- and extracellular levels are not directly comparable to the *in vivo* situation and that secretion defects are likely to be even more pronounced in the patients' cells.

One could speculate that the activity of ADAMTS13 at the endothelial surface would be the most important in patients, since it is the site of the most active VWF multimers, especially, under conditions that trigger USS episodes. We showed here, that almost all investigated ADAMTS13 variants exhibit considerable residual activity under flow conditions in the string assay at concentrations as low as 100 ng/ml. Our data further indicate that the same amount would also suffice to proteolize VWF-platelet complexes. It is thus intriguing to suggest an approach, which increases ADAMTS13 secretion as a novel therapeutic option for USS.

Such a strategy is commonly used to treat milder types 1 and 2A of von Willebrand disease [58]. For some VWF variants with impaired secretion, desmopressin is employed as a therapeutic agent to induce release of the partially functional variants. Of course, desmopressin itself must not be used to increase ADAMTS13 secretion, as it would further increase the already high VWF plasma concentration during a TTP bout. Thus, finding a suitable substance and development of such an approach require thorough investigation and understanding of the secretion mechanism of ADAMTS13 and the possibilities of its regulation in the future.

## Supporting information

**S1 Fig. Multimer analysis of recombinant wtVWF.** Multimer analysis of pooled plasma (left lane) and recombinant wtVWF (right lane) was performed by SDS–agarose gel electrophoresis and immunoblotting onto a nitrocellulose membrane with luminescent visualization. The figure is composed of one gel. The black line indicates deleted lanes with multimers not relevant for this study.
(PDF)

**S2 Fig.** Magnified ROIs of panels (A) and (B) are shown in (C) and (D), respectively. To show the VWF strings, the yellow lines were moved below the strings (above for strings 4, 49 and 63) and the beginning of the strings is marked by red arrows. Strings 13 and 26, which were not cleaved after 10.5 min are marked by a blue arrow (D).
(PDF)

**S3 Fig. Degradation quantification via changepoint analysis using binary segmentation.** Three representative quantification visualizations of LTA measurements summarized in Fig 4 are shown. The raw signal is depicted in black and the intra-changepoint mean between two consecutive changepoints is represented by horizontal red line segments. Each signal is sampled at two points in time (8 min and 50 min; blue vertical line segments) by calculating the intra-changepoint mean at this point (point of intersection of blue and red line segment). The

difference between these two means quantifies the magnitude of degradation.
(PDF)

## Author Contributions

**Formal analysis:** Anton Letzer, Katja Lehmann, Christian Mess, Stefan W. Schneider.

**Investigation:** Anton Letzer, Katja Lehmann, Gesa König, Tobias Obser, Sonja Schneppenheim, Ulrich Budde, Maria A. Brehm.

**Methodology:** Sonja Schneppenheim, Ulrich Budde, Maria A. Brehm.

**Resources:** Sven Peine, Reinhard Schneppenheim.

**Supervision:** Maria A. Brehm.

**Writing – original draft:** Maria A. Brehm.

**Writing – review & editing:** Ulrich Budde, Stefan W. Schneider, Reinhard Schneppenheim.

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
