## [Decision Letter · Decision Letter 0]

6 Mar 2020

PONE-D-20-00868

Upshaw-Schulman syndrome-associated ADAMTS13 variants possess proteolytic activity at the surface of endothelial cells and in simulated circulation

PLOS ONE

Dear Dr Brehm,

Thank you for submitting your manuscript to PLOS ONE. After careful consideration, we feel that it has merit but does not fully meet PLOS ONE’s publication criteria as it currently stands. Therefore, we invite you to submit a revised version of the manuscript that addresses the points raised during the review process.

The manuscript has been reviewed by two experts in the field. They recommend to provide several revisions. The authors should indicate the number of repetitions in the different experiments, including the measurements of protein concentration in immunoblots. Where those experiments performed once? 

Additionally, it is not clear how the lowest producing mutants were obtained in order to have similar concentration as the wild type protein in the functional assays. 

Several spelling and grammar errors should be corrected.

We would appreciate receiving your revised manuscript by Apr 20 2020 11:59PM. To enhance the reproducibility of your results, we recommend that if applicable you deposit your laboratory protocols in protocols.io, where a protocol can be assigned its own identifier (DOI) such that it can be cited independently in the future. For instructions see: http://journals.plos.org/plosone/s/submission-guidelines#loc-laboratory-protocols

We look forward to receiving your revised manuscript.

Kind regards,

Pablo Garcia de Frutos

Academic Editor

PLOS ONE

Journal Requirements:

"This study was partially financially supported by research funding from the German Research Foundation (DFG) to the Research Group FOR1543, specifically authors R.S., M.A.B., T.O., G.K., U.B., Grant number: SCHN325/7-2). "

We note that one or more of the authors are employed by a commercial company: MEDILYS Laborgesellschaft mbH.

Reviewers' comments:

Reviewer's Responses to Questions

**Comments to the Author**

1. Is the manuscript technically sound, and do the data support the conclusions?

Reviewer #1: Yes

Reviewer #2: Yes

2. Has the statistical analysis been performed appropriately and rigorously? 

Reviewer #1: N/A

Reviewer #2: I Don't Know

3. Have the authors made all data underlying the findings in their manuscript fully available?

Reviewer #1: Yes

Reviewer #2: Yes

4. Is the manuscript presented in an intelligible fashion and written in standard English?

Reviewer #1: Yes

Reviewer #2: Yes

5. Review Comments to the Author

Reviewer #1: This is a very interesting manuscript describing the activity of 9 ADAMTS13 mutation from Upshaw Schulman syndrome patients using flow assays

The data is pretty clear and well presented

I had only have minor comments

Although there is already a publication describing the assay, pictures of the string assay would be useful to improve the readability of the manuscript.

The statement on line 308 "surprisingly, all of the examined variants exhibited almost the same activity as WtADAMTS13 after 10 min. ..." is very misleading as this is likely the results of some form of saturation and is in fact a limitation of the assay.

Authors acknowledge that in the following graph and in the figures, but the statement is incorrect and should be removed.

The variant ADAMTS13 appear to have residual activity but not as strong as the wt proteins.

Reviewer #2: The study describes the activity of different ADAMTS13 mutants associated with Upshaw-Schulman syndrome. The authors provide novel and interesting methodology to test different aspects of the activity of this complex molecule, providing new insights on the VWF degradation pathway that underlies the syndrome. Therefore, the subject is of interest in the field.

There are several aspects in the presentation of the results that should be explained in more detail. The presentation of the results have several repetitions. The distribution of the figures should be changed in order to present the data generated by the same experiments in a more homogeneous fashion. Finally, the text needs several spelling and format corrections, some of them indicated below.

Table 1. The recombinant expression and static measurements seems to have been performed once. Therefore, it is difficult to establish minor differences in expression/secretion. The authors should mention this shortcoming in the text.

The authors state that the static ADAMTS13 measurements are done on “equal concentrations of all recombinant proteins” (L286). This is difficult to understand for the case of the variants with lowest concentration (for instance, pL232Q), as this would mean that the authors concentrate more than 100 times more their media compared to wtADAMTS13 media. Is this correct? Could the authors be more clear on the protocol employed?

Table 2 repeats the same data of Table 1. Eliminate Table 1. Add the SEM when available.

Figure 3 presents data from figure 2, as mentioned in the figure legend. Therefore, the data will be better presented as two sections of a single figure.

Figure 4. The experiment is difficult to interpret, as before the addition of the different ADAMTS13 the trace of the curves differs. This is especially evident in the case of no ADAMTS13 addition. The authors provide a quantification of the results in the next two graphs. However, Figure 5 represents an example of the calculations and would be better presented as a supplementary information. Figure 6 should be combined with Figure 4.

Figure 7. Provide a quantification of the experiment as in figure 6.

L49. Could the authors provide a recent review on ADAMTS13?

L108. Remove the location of the company’s product, as this is not provided in all cases and do not always represent the origin of the product.

L111. “Four μg…” instead of “4 μg…”

L116 “1%” (similar in L136, L139…)

L117. “Seventy-two hours…” instead of “72 hours…” (similar in L145).

L134. “4 oC” (similar in L136; L168; L171)

L140. The authors use a very fast incubation of less than 5 minutes. This could produce inconsistencies in the incubation time of the different wells of the plate. Do the authors account for that?

L142 Could the authors specify the meaning of “anthos”?

L162 gelatin?

L166 “and” instead of “an”

L285 change “to receive” to “to obtain”

L368 “Exemplary analysis” change it to “A representative result”

L413 To determine the minimal concentration of ADAMTS13 required…

6. PLOS authors have the option to publish the peer review history of their article (what does this mean?). If published, this will include your full peer review and any attached files.

Reviewer #1: No

Reviewer #2: No

---

## [Author Response · Author response to Decision Letter 0]

9 Apr 2020

The Responses to the Reviewers are also provided in the attached document "Response to Reviewers"

Reviewer #1: This is a very interesting manuscript describing the activity of 9 ADAMTS13 mutation from Upshaw Schulman syndrome patients using flow assays

The data is pretty clear and well presented.

I had only have minor comments

Although there is already a publication describing the assay, pictures of the string assay would be useful to improve the readability of the manuscript.

Authors: Exemplary, representative images of the string assay and the string measurements at time points 0 and 10.5 min for wtADAMTS13 are now provided in Supplemental Figure S2, file name: S2_Fig.pdf.

Reviewer #1: The statement on line 308 "surprisingly, all of the examined variants exhibited almost the same activity as WtADAMTS13 after 10 min. ..." is very misleading as this is likely the results of some form of saturation and is in fact a limitation of the assay.

Authors acknowledge that in the following graph and in the figures, but the statement is incorrect and should be removed.

The variant ADAMTS13 appear to have residual activity but not as strong as the wt proteins.

Authors: The sentence in line 335 has been modified accordingly. Additionally, the limitation of the assay was further explained in the Discussion line 460:” Since a limited amount of strings is available for cleavage, which is probably lower compared to the in vivo situation, this observation indicates a limitation of this assay that might explain the higher residual activity of all ADAMTS13 variants in the string cleavage assay..”.

Reviewer #2: The study describes the activity of different ADAMTS13 mutants associated with Upshaw-Schulman syndrome. The authors provide novel and interesting methodology to test different aspects of the activity of this complex molecule, providing new insights on the VWF degradation pathway that underlies the syndrome. Therefore, the subject is of interest in the field.

There are several aspects in the presentation of the results that should be explained in more detail. The presentation of the results have several repetitions. The distribution of the figures should be changed in order to present the data generated by the same experiments in a more homogeneous fashion. Finally, the text needs several spelling and format corrections, some of them indicated below. 

Authors: Spelling and grammar corrections, additionally to the ones mentioned below, were made throughout the manuscript and marked in red.

Reviewer #2: Table 1. The recombinant expression and static measurements seems to have been performed once. Therefore, it is difficult to establish minor differences in expression/secretion. The authors should mention this shortcoming in the text.

Authors: That the Western blot and static measurements were done from one protein batch is now mentioned in Table Legend 1 (line 279). Furthermore, the following has been added to the discussion, starting line 485: “We also observed reduced secretion for all variants. Since we have performed this experiment only with one batch of stably expressed proteins, we cannot rule out minor differences in expression and secretion between different protein expressions. Nonetheless, the data shown here are in good accordance with our previous study, investigating secretion after transient transfection [30]. Compared to the stable expression shown here, then, only p.Leu232Gln exhibited higher intracellular and extracellular levels and p.Asp235Tyr showed a slightly decreased secretion. These data indicate that some differences in the extent of secretion defects can be observed between stable and transient recombinant protein expressions”.

Reviewer #2: The authors state that the static ADAMTS13 measurements are done on “equal concentrations of all recombinant proteins” (L286). This is difficult to understand for the case of the variants with lowest concentration (for instance, pL232Q), as this would mean that the authors concentrate more than 100 times more their media compared to wtADAMTS13 media. Is this correct? Could the authors be more clear on the protocol employed?

Authors: The missing information was added lines 122-127: “All ADAMTS13 variants were concentrated 20-fold. Afterwards the yielded concentration was determined by the Imubind® ADAMTS13 ELISA (Sekisui Diagnostics). To yield a concentration >3000 ng/ml for all mutants, the required factor for further concentration was estimated for low expressing variants, which then underwent one additional concentration step. In total, low expressing variants were concentrated between 50 to 200-fold”.

Reviewer #2: Table 2 repeats the same data of Table 1. Eliminate Table 1. Add the SEM when available.

Authors: Table 1 was deleted, and available SD and SEM were added to Table 2 (now Table 1).

Reviewer #2: Figure 3 presents data from figure 2, as mentioned in the figure legend. Therefore, the data will be better presented as two sections of a single figure.

Authors: The two figures were combined to new Figure 2.

Reviewer #2: Figure 4. The experiment is difficult to interpret, as before the addition of the different ADAMTS13 the trace of the curves differs. This is especially evident in the case of no ADAMTS13 addition. 

Authors: To address these differences between the curves, we have used the employed changepoint analysis, which quantified the differences between the plateaus before and after addition of ADAMTS13 for each single curve, as visualized in exemplary analyses shown in Figure 5 (now Supplemental Figure S3, File S3_Fig.pdf).

Reviewer #2: The authors provide a quantification of the results in the next two graphs. However, Figure 5 represents an example of the calculations and would be better presented as a supplementary information. 

Authors: Figure 5 was moved to the Supplemental Data (now Supplemental Figure S3, file S3_fig.pdf)

Reviewer #2: Figure 6 should be combined with Figure 4.

Authors: The two figures were combined to new Figure 3.

Reviewer #2: Figure 7. Provide a quantification of the experiment as in figure 6.

Authors: The quantification is now shown in Figure 4B

Reviewer #2:L49. Could the authors provide a recent review on ADAMTS13?

Authors: A recent review was added to the references (new reference 5: South K, Lane DA. ADAMTS-13 and von Willebrand factor: a dynamic duo. Journal of thrombosis and haemostasis : JTH. 2018;16(1):6-18.)

Reviewer #2: L108. Remove the location of the company’s product, as this is not provided in all cases and do not always represent the origin of the product. 

Authors: Company locations were deleted throughout the manuscript.

Reviewer #2:L111. “Four μg…” instead of “4 μg…”

Authors: Changed, now line 110.

Reviewer #2:L116 “1%” (similar in L136, L139…)

Authors: Spaces between numbers and % were deleted throughout the manuscript.

Reviewer #2:L117. “Seventy-two hours…” instead of “72 hours…” (similar in L145).

Authors: Changed, lines 117, 153.

Reviewer #2: L134. “4 oC” (similar in L136; L168; L171) 

Authors: Corrected, now lines 139, 142, 180, 182.

Reviewer #2: L140. The authors use a very fast incubation of less than 5 minutes. This could produce inconsistencies in the incubation time of the different wells of the plate. Do the authors account for that?

Authors: Actually, this was a copy paste error from our general protocol. Thank you for pointing out this mistake. The incubation time was corrected to 5 min, line 148. 

To keep the incubations times as equal as possible in the different wells, we added the substrate employing a 12-channel pipet to speed up this step, incubated 5 min and then the stopping reagent was added with the same pipet and similar speed. 

Reviewer #2:L142 Could the authors specify the meaning of “anthos”? 

Authors: Anthos is the company. Changed to “microplate reader htIII (anthos)”, line 150.

Reviewer #2:L162 gelatin? 

Authors: Corrected to gelatine line 175.

Reviewer #2:L166 “and” instead of “an” 

Authors: Corrected, line 179.

Reviewer #2: L285 change “to receive” to “to obtain” 

Authors: Changed, line 314.

Reviewer #2: L368 “Exemplary analysis” change it to “A representative result”

Authors: Since the respective figure was moved to the supplementals, changed to: “Changepoint analysis (examples shown in S3 Fig)…”, line 377.

Reviewer #2: L413 To determine the minimal concentration of ADAMTS13 required…

Authors: The Figure title and legend were adjusted, lines 415-428.

---

## [Editor Report · Decision Letter 1]

20 Apr 2020

Upshaw-Schulman syndrome-associated ADAMTS13 variants possess proteolytic activity at the surface of endothelial cells and in simulated circulation

PONE-D-20-00868R1

Dear Dr. Brehm,

We are pleased to inform you that your manuscript has been judged scientifically suitable for publication and will be formally accepted for publication once it complies with all outstanding technical requirements.

With kind regards,

Pablo Garcia de Frutos

Academic Editor

PLOS ONE
---

## [Editor Report · Acceptance letter]

24 Apr 2020

PONE-D-20-00868R1 

Upshaw-Schulman syndrome-associated ADAMTS13 variants possess proteolytic activity at the surface of endothelial cells and in simulated circulation 

Dear Dr. Brehm:

I am pleased to inform you that your manuscript has been deemed suitable for publication in PLOS ONE. Congratulations! Your manuscript is now with our production department. 

With kind regards,

on behalf of

Dr. Pablo Garcia de Frutos 

Academic Editor

PLOS ONE